# Variations in Solar Radiation and Their Effects on Rice Growth in Agro-Photovoltaics System

**Yamin Jia [1], Xiaoli Gao [1,\*], Junkang He [2], Jiufu Luo [3], Xin Sui [3] and Peilan Su [1]**

[1]   College of Water Resources Sciences and Engineering, Taiyuan University of Technology, Taiyuan 030024, China; jiayamin@tyut.edu.cn (Y.J.); supeilan@tyut.edu.cn (P.S.)

[2]   College of Horticulture, Shenyang Agricultural University, Shenyang 110866, China; hehuachun@tyut.edu.cn

[3]   China Institute of Water Resources and Hydropower Research, Beijing 100038, China; luojf@iwhr.com (J.L.); suixin@iwhr.com (X.S.)

\*   Correspondence: gaoxiaoli01@tyut.edu.cn

## Abstract

Agro-photovoltaics (APV) or agrivoltaic systems integrate crop cultivation with solar energy production, offering a promising solution through the dual-use of land. This two-year study (2023 and 2024) examined the effects of an APV system on rice production. The results indicated that APV arrays created spatially variable light environments, with shadow lengths following predictable solar azimuth patterns and cloudy conditions mitigating shading effects through enhanced diffuse light. Compared with CK (non-shadow area), inter-panel plots (BP) maintained 77% photosynthetic efficiency and 85.4% plant height, whereas the areas beneath the panel showed a significant decrease in the relative chlorophyll content (SPAD values), photosynthesis rates, and yield. BP plots preserved a 78% fruiting rate through adaptive stomatal regulation, whereas LP zones (directly under the low eave) exhibited 35% higher intercellular $CO_2$ because of the limited assimilation in shading. Rice yield losses were correlated with shading intensity, driven by reduced panicles and grain filling. Moreover, the APV system achieved a high land equivalent ratio of 148–149% by combining 65–66% rice yield with 82.5% photovoltaics output. Based on the microenvironment created by the APV system, optimal crop types and fertilisation are essential for enhancing agricultural yields and improving land use efficiency.

**Keywords:** agrivoltaic system; sunshine hours; photosynthetic property; land equivalent ratio

## 1. Introduction

The integration of photovoltaics (PV) systems with agricultural production, known as agro-photovoltaics (APV), represents a promising strategy to address the dual challenges of renewable energy generation and sustainable land use [1,2]. As global energy demand continues to increase, the deployment of large-scale solar farms has increased competition for arable land, particularly in regions with high solar insolation [3,4]. Converting even <1% of cropland to agrivoltaic systems is estimated to offset the global energy demand [5]. There are two types of APV systems: pre-existing PV array or intentionally designed APV systems [6]. APV systems offer a synergistic solution by co-locating solar panels with crop cultivation, thereby optimising land use efficiency and potentially enhancing ecosystem services, such as water conservation and microclimate regulation [7,8]. However, microclimatic heterogeneities and their effect on crop production are complex and variable [9]. The success of APV systems relies on an understanding of the trade-offs between energy

production and agricultural productivity, particularly for staple crops, such as rice, which are highly sensitive to light availability [10,11].

Recent studies have systematically examined the effects of APV shading on various crops, revealing complex opportunities and challenges. Partial shading reduces evapotranspiration and soil moisture loss, which results in significant agronomic benefits in arid or semi-arid regions under APV panels [2,12]. However, the reduction in photosynthetically active radiation beneath the PV arrays can result in decreased biomass accumulation and yield, particularly for light-sensitive crops, such as maize and rice. Studies on rice cultivation under APV systems have shown yield reductions ranging from 13% to 30%, depending on shading intensity and panel configuration [11,13]. These results highlight the need for site-specific optimisation of APV designs to balance energy output and agricultural productivity. Despite the current advances, gaps remain in understanding how dynamic shading patterns under APV systems affect rice growth physiology, particularly with respect to photosynthetic efficiency and yield.

Rice has long been a crucial agricultural crop in China, particularly in the southern region. Consequently, interest in APV in southern China has resulted in the need to examine its effect on rice growth and yield. This study aims to investigate spatiotemporal variations in solar radiation beneath APV arrays and their effects on rice growth, photosynthesis, and yield components. Specifically, we seek to (1) characterise the shading patterns and photosynthetic photon flux density (PPFD) distribution under an APV array; (2) evaluate the effects of different light regimens on the growth parameters and photosynthetic characteristics of rice; and (3) analyse yield components for varying light conditions. This study provides insights for optimising APV system designs while maximising land use efficiency in rice-growing regions.

## 2. Materials and Methods

### 2.1. Study Area and Site Conditions

These experiments were conducted from June 2023 to October 2023 and June 2024 to October 2024 at the 99 MW agrivoltaic project in Qizhou Town, Qichun County, China (30°0′23.64″ N, 115°23′15.37″ E). The site was located in the northern subtropical monsoon climate zone of the middle-lower Yangtze River basin. The agrivoltaic arrays cover an area of 140 hectares, at an elevation of 24 m above sea level. The site receives abundant solar radiation with 2025.8 sunshine hours annually, which classifies it as a solar-rich resource area [14].

The major local soil type is a yellowish clay loam soil exhibiting favourable agricultural properties, including a slightly acidic pH (5.8–6.3), 3% organic matter, and a cation exchange capacity of 18 cmol(+)/kg, indicating a strong nutrient retention capacity. Before beginning the agrivoltaic project, the land was primarily used for rice cultivation and represented typical agricultural land use in this subtropical paddy field ecosystem. The combination of high availability of solar resource and fertile soil conditions makes the site particularly suitable for evaluating the dual-use potential of agrivoltaic systems in the intensive agricultural regions of China.

The daily temperature and rainfall data for the rice-growing period in 2023 (A) and 2024 (B) are shown in Figure 1. The highest and lowest average monthly temperatures were recorded in July (29.03 °C) and October (19.90 °C) in 2023 and in August (31.17 °C) and October (19.73 °C) in 2024. The precipitation in 2023 and 2024 was 676.8 mm and 635.6 mm, respectively. Additionally, the total sunshine hours during the rice-growing period in 2023 and 2024 were 1010 h and 890 h, respectively.

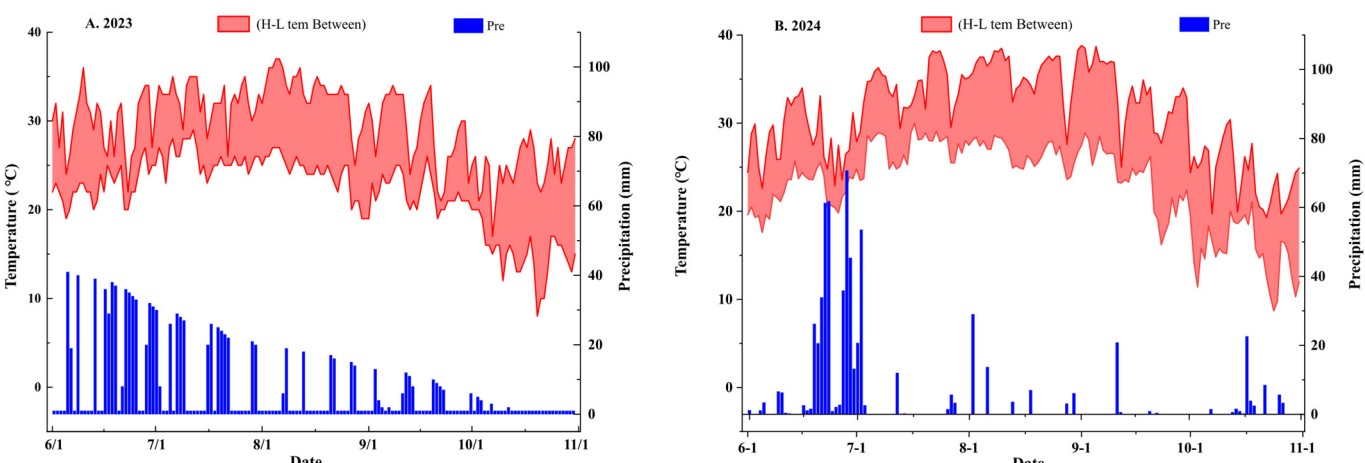

**Figure 1.** The highest and lowest daily temperature and precipitation during the rice growth period in Qizhou Town. Note: "H-L tem Between" means the daily temperature change scale between the highest and the lowest; "Pre" is the abbreviation of precipitation.

*2.2. APV Array and Experiment Design*

The experimental area featured fixed-tilt, bifacial PV arrays arranged in north–south oriented rows with 7.5 m spacing between adjacent racking centres. The PV silicon panels were maintained at a fixed tilt angle of 28°, with the front (southern) eave positioned at a vertical height of 2.5 m and the rear (northern) eave 4.5 m above ground level. The agricultural cultivation zones were established in three distinct positions relative to the APV arrays, as shown in Figure 2: (1) HP plots, directly beneath the high edge (rear side of the southern APV panels), defined by the vertical height of the 4.5 m edge, the plot width was 2.0 m; (2) LP plots, directly under the low edge (front side of the northern APV panels) corresponding to the 2.5 m edge height, the plot width was 2.0 m; and (3) BP plots, the inter-row space between the southern and northern arrays, the plot width was 3.5 m. An adjacent APV system with full sun was used as the control plot (CK). This configuration provided four distinct microenvironments for the comparative analysis of crop performance under varying light conditions, while maintaining uniform soil management practices across all treatments.

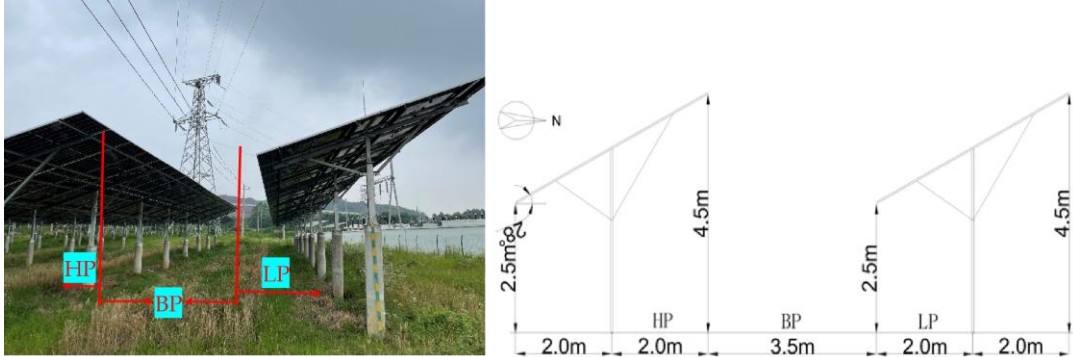

**Figure 2.** Agro-photovoltaics (APV) array and experiment plots in the study site. HP plots were the area directly beneath the high edge of the APV panels to the bases of the panels. BP plots were the inter-row space between the southern and northern APV panels. LP plots were the area directly beneath the low edge of the APV panels to the bases of the panels.

To maximise the available cultivation area within the agrivoltaic system, the planting zones were systematically designed according to the fixed APV array dimensions in north–south and east–west orientations. The HP and LP cultivation areas each measured

40 m in length and 2 m in width, thus creating individual plot sizes of 80 m². The BP zone spanned 40 m in length with a 3.5 m width, providing 140 m² of cultivation space per plot. To ensure statistical validity, each treatment (HP, LP, BP, and CK) was replicated three times, resulting in a total of 12 experimental plots (4 treatments × 3 replicates) distributed in the study area.

The experiment was conducted using the indica hybrid rice cultivar Liangyou 1928, a two-line hybrid variety characterised by high yield, superior grain quality, and enhanced stress resistance. The rice cultivation trials were performed during identical growing seasons in 2023 and 2024, with sowing on 6 June and harvesting on 21 October, corresponding to a 135-day complete growth cycle. The direct seeding method was used following standardised field preparation, including fertilisation, puddling, rotary tillage, levelling, and drainage. Crop management followed the following regional best practices: shallow flooding (3–5 cm) during seedling emergence, thin water layer maintenance during tillering, field drying at the maximum tillering stage, deeper flooding (8–10 cm) at panicle initiation, irrigation maintained during grain filling, and moist soil conditions during grain maturation. Fertilisation consisted of nitrogen, phosphorus, and potassium fertiliser applied at 450 kg/ha based on standard local protocols.

### 2.3. Evaluation of PV Panel Layout on Crop Sunlight Exposure

### 2.3.1. Calculation of Solar Declination, Altitude Angle, and Azimuth Angle

The solar declination varies seasonally, reaching 0° during the spring and autumn equinox, and peaking at 23°23′ during the summer and winter solstice. According to Shan et al. [15], the solar declination ($\delta$), altitude angle ($\alpha$), and azimuth angle ($\beta$) were calculated as follows:

$$\delta = 0.3730 + 23.2567\sin\theta + 0.1149\sin2\theta - 0.1717\sin3\theta - 0.7580\cos\theta + 0.3650\cos2\theta + 0.0201\cos3\theta \tag{1}$$

where $\theta$ is the solar position angle, $\theta = 2\pi t/365.2422$, $t = N - N_0$, and N is the day of the year, $N_0 = 79.6764 + 0.2422 \times (\text{Year} - 1985) - \text{INT}[(\text{Year} - 1985)/4]$.

$$\text{Sin}\alpha = \sin\varphi\sin\delta + \cos\varphi\cos\delta\cos\omega \tag{2}$$

where $\omega$ is the hour angle, $\omega = 15°(t - 12)$, forenoon (08:00–12:00) is a negative value, and afternoon (12:00–16:00) is a positive value.

$$\text{Cos}\beta = (\sin\alpha\sin\varphi - \sin\delta) \cos\alpha\cos\varphi \tag{3}$$

### 2.3.2. Calculation of Shadow Length of PV Panel and Effective Shadow Length

The shadow length of the PV panels (L) on the ground refers to the length of the shadow cast by the PV panels resulting from sunlight obstruction. It depends on the solar altitude angle ($\alpha$) and the height of the PV (H), which is calculated as follows:

$$L = H/\tan\alpha \tag{4}$$

where L is the shadow length of the PV panels, H is the height of the PV panels, and $\alpha$ is the solar altitude angle.

Effective shadow length (L′) refers to the shadow cast by PV panels along the north–south direction, which creates functional shading for crops planted between the panel rows. It may be calculated as follows:

$$L' = \begin{cases} L\cos\beta, & 0 \leq \beta < \pi/2 \\ L\cos(\pi - \beta), & \pi/2 \leq \beta < \pi \\ L\cos(\beta - \pi), & \pi \leq \beta < 3\pi/2 \\ L\cos(2\pi - \beta), & 3\pi/2 \leq \beta < 2\pi \end{cases} \tag{5}$$

where L′ is the effective shadow length, L is the shadow length of the PV panels, and β is the azimuth angle.

### 2.3.3. Measurement of Light Intensity in Agrivoltaic System

The PPFD in the agrivoltaic field was measured using a TES-1399P illuminometer (TES Electrical Electronic Corp, Taiwan, China). During the rice-growing period from 15 June to 21 October, daylight intensity measurements were collected under three characteristic weather conditions: sunny, cloudy, and rainy days. For each hourly measurement interval, five sampling points were established within each treatment plot, with the arithmetic mean of the measurements recorded as the representative light intensity value for that specific treatment and time.

### 2.4. Rice Growth, Photosynthetics, and Yield

Rice progresses through distinct developmental phases from seed germination to maturity, encompassing root establishment, leaf expansion, tillering, stem elongation (jointing), panicle formation, flowering, grain filling, and final maturation. This study examined four critical growth stages—seedling emergence, tillering, stem elongation–panicle initiation, and heading–grain filling—to evaluate growth patterns and photosynthetic performance. Plant height was determined by measuring the vertical distance from the soil surface to the tallest leaf tip of three randomly selected plants per plot using a measuring tape, with mean values calculated for each treatment. Tillering capacity was assessed during the tillering stage by counting tillers from three representative plants per plot. SPAD values (the relative chlorophyll content) of the fully expanded penultimate leaf was measured with a SPAD-502 chlorophyll metre (five readings per leaf, averaged). Photosynthetic parameters—including net photosynthetic rate (Pn), transpiration rate (Tr), stomatal conductance (Gs), and intercellular $CO_2$ concentration (Ci)—were monitored at 10:00, 12:00, 14:00, and 16:00 on sunny and clear days using the Li-6400 (Lincoln, NE, USA) photosynthesis system. Five readings were taken on each leaf and averaged over three replicate plants per plot.

At physiological maturity, the grain yield and its components were quantified within 1 m$^2$ sampling areas for all treatments. Yield parameters included (1) the productive panicle density (panicles/m$^2$), determined by manual counting; (2) the seed-setting rate (%), calculated as (number of filled grains/total florets × 100); (3) the grain weight per panicle (g), measured by oven-drying at 80 °C until constant weight, and weighing all filled grains from the sampled panicles; and (4) the 1000-grain weight (g), determined from three replicates of randomly selected filled grains.

### 2.5. Land Equivalent Ratio

Based on the land equivalent ratio (LER) calculation equation reported by Trommsdorff et al. [16], we calculated and evaluated the land use efficiency of the agrivoltaic system per unit area relative to single systems (pure agriculture or pure PVs). LER is calculated as follows:

$$LER = \frac{Yield\ a\ (dual)}{Yield\ a\ (mono)} + \frac{Yield\ b\ (dual)}{Yield\ b\ (mono)} \tag{6}$$

where *a* and *b* are the production of crop and electricity generating capacity, respectively, dual is the environment of the agrivoltaic system, and mono is the environment of pure agriculture or PVs.

*2.6. Statistical Analysis*

All experimental data were recorded and processed using Microsoft Excel 2016, analysed for statistical significance with SPSS 25, and visualised using Origin 2025 for graphical representations. ANOVA was conducted to determine the difference among four plots on measurement data at a probability level of 0.05.

## 3. Results

*3.1. Effect of APV Array Shading on Daily Variations in Direct Sunlight Exposure Duration*

The solar altitude angle, azimuth angle, and effective shadow length caused by the PV panels in the agrivoltaic system on 15 June 2023, as influenced by seasonal and diurnal variations, are listed in Table 1. During mid-June, the solar altitude angle gradually increased from 8.2° at 06:00 to its peak of 83.1° at 12:00 and progressively decreased to 8.7° by 18:00. Concurrently, the solar azimuth angle showed distinct directional changes. At sunrise, the sun was positioned in the northeast at 61.3°, shifted to due east (90.1°) by around 09:00, and reached due south (180.5°) by noon. In the afternoon, the sun continued its westward path, arriving at due west (approximately 270°) near 16:00 before moving towards the northwest and reaching 298.4° relative to true north by 18:00.

**Table 1.** Solar altitude angle, azimuth, and effective shadow length on 15 June 2023 at study site.

| Time | Solar Declination (°) | Longitude (°) | Solar Altitude Angle (°) | Solar Azimuth Angle (°) | Effective Shadow Length (m) |
|---|---|---|---|---|---|
| 6:00 | 23.33 | 30.15 | 8.2 | 61.3 | 15.00 |
| 7:00 | 23.33 | 30.15 | 20.1 | 74.5 | 3.29 |
| 8:00 | 23.33 | 30.15 | 33.8 | 84.2 | 0.68 |
| 9:00 | 23.33 | 30.15 | 48.3 | 90.1 | 2.15 |
| 10:00 | 23.33 | 30.15 | 62.7 | 109.8 | 2.29 |
| 11:00 | 23.33 | 30.15 | 75.2 | 129.5 | 0.91 |
| 12:00 | 23.33 | 30.15 | 83.1 | 180.5 | 0.54 |
| 13:00 | 23.33 | 30.15 | 75.4 | 230.3 | 0.75 |
| 14:00 | 23.33 | 30.15 | 63.0 | 255.1 | 0.59 |
| 15:00 | 23.33 | 30.15 | 48.7 | 260.3 | 0.67 |
| 16:00 | 23.33 | 30.15 | 34.2 | 275.5 | 0.63 |
| 17:00 | 23.33 | 30.15 | 20.5 | 285.2 | 3.16 |
| 18:00 | 23.33 | 30.15 | 8.7 | 298.4 | 13.99 |

The effective shadow length cast by the PV panel array showed systematic variations because of the combined effect of the solar altitude and azimuth angles. At 06:00, with the sun positioned in the northeast at a low altitude angle, the shadow length reached approximately 15.00 m. As the day progressed, the increased solar altitude angle and shifting azimuth angle caused the shadow length to decrease rapidly. By 08:00, when the sun was due east, the effective shadow length decreased to 0.68 m. At solar noon (12:00), with the sun directly south at its highest elevation, the shadow length was minimised to 0.54 m. In the afternoon, the shadow length followed a symmetrical pattern similar to that in the morning, gradually increasing as the sun descended, reaching 13.99 m by 18:00.

Using the time points from Table 1 for the x-axis and the corresponding effective shadow lengths of the PV array for the y-axis, a point–line graph (Figure 3) was plotted. As shown in Figure 3, the effective shadow length exhibited monotonic variations across four distinct azimuth angle intervals: 0–90°, 90–180°, 180–270°, and 270–360°.

Furtherly, curve fitting of the effective shadow length over time points was performed at each monotonic interval. The resulting fitted curves are listed in Figure 3.

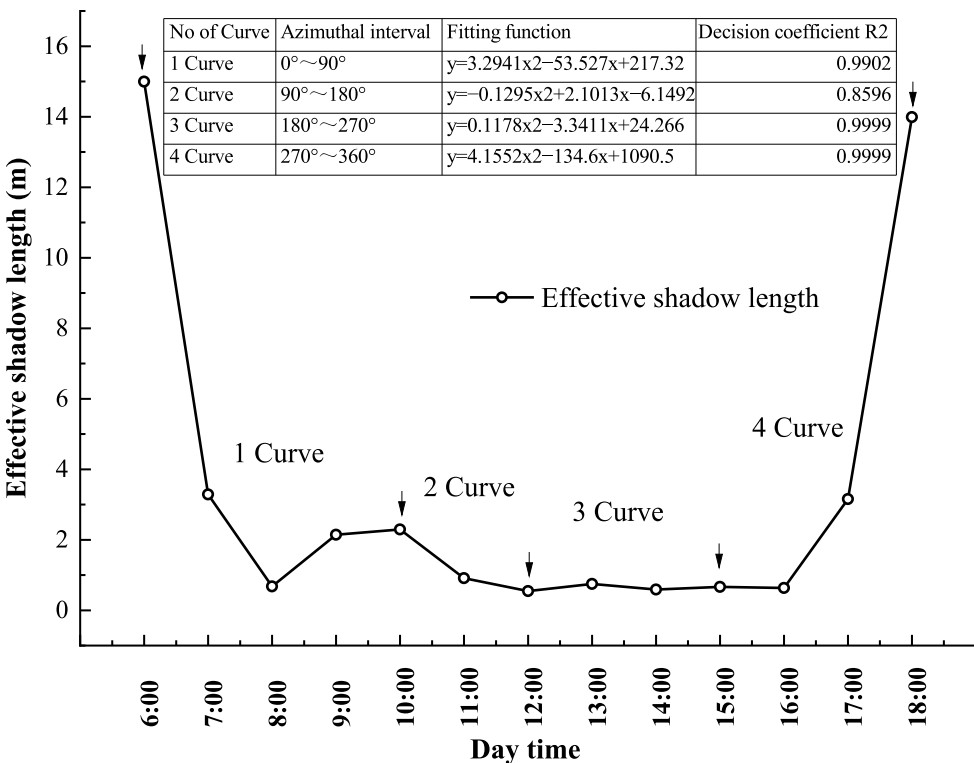

| No of Curve | Azimuthal interval | Fitting function | Decision coefficient R2 |
|---|---|---|---|
| 1 Curve | 0°~90° | y=3.2941x2−53.527x+217.32 | 0.9902 |
| 2 Curve | 90°~180° | y=−0.1295x2+2.1013x−6.1492 | 0.8596 |
| 3 Curve | 180°~270° | y=0.1178x2−3.3411x+24.266 | 0.9999 |
| 4 Curve | 270°~360° | y=4.1552x2−134.6x+1090.5 | 0.9999 |

**Figure 3.** Daily variation in the effective shadow length of photovoltaics (PV) panels and fitting curve function in the APV area on 15 June 2023 at the study site.

### 3.2. Changes in Daylight Duration and PPFD Under APV Array During Rice Growth Period

Based on the fitted curves of shadow lengths across different solar angle intervals (Figure 3), the initial sunshine times for the various rice planting plots within the PV array area can be determined. Consequently, the number of sunshine hours can be determined. The number of sunshine hours for each rice cultivation treatment in the agricultural and PV complementary area is listed in Table 2, combined with the solar orientation and the shadow length of the PV array on 15 June 2023. As listed in Table 2, the sequence of sunshine hours on 15 June was as follows: CK > BP > HP > LP. Compared with the control group, a decrease of 10.80% in the number of sunshine hours per day was observed between the PV array area. Moreover, a significant decrease of 84.37% and 97.56% in the number of sunshine hours per day was observed in the front and back of the PV panels, respectively. The period of light availability was limited to a shorter duration after sunrise or before sunset.

**Table 2.** Sunshine duration across different rice planting plots in agro-photovoltaics field on 15 June 2023.

| Plot | Azimuth: 0–90° | | Azimuth: 90–180° | | Azimuth: 180–360° | | Total Sunshine Hours |
|---|---|---|---|---|---|---|---|
| | Boundary Light Onset Moment | Boundary Light End Moment | Boundary Light Onset Moment | Boundary Light End Moment | Boundary Light Onset Moment | Boundary Light End Moment | |
| CK | 6.00 | 9.00 | 9.00 | 12.00 | 12.00 | 19.30 | 13.50 |
| LP | 7.96 | 8.29 | - | - | - | - | 0.33 |
| BP | 7.07 | 9.17 | 9.20 | 15.37 | 17.02 | 19.16 | 9.25 |
| HP | - | - | - | - | 15.14 | 17.25 | 2.11 |

Note: The plot labelled CK was the control plot, which was the not shaded area of the rice plot. LP was the area directly beneath the low edge of the APV panels to the bases of the panels. BP was the inter-row space between the southern and northern APV panels. HP was the area directly beneath the high edge of the APV panels to the bases of the panels.

To clarify the dynamics of daily sunshine hours during the rice growth period, daily sunshine hours were calculated for each plot on 15 July, 15 August, 15 September, and 15 October 2023. The results are shown in Figure 4A. During the rice growth period, the

daily sunshine hours for the CK plot were 11.3–14.1 h, 1.6–2.1 h for the HP plot, 7.7–9.5 h for the BP plot, and for the 0.27–0.33 h LP plot approximately.

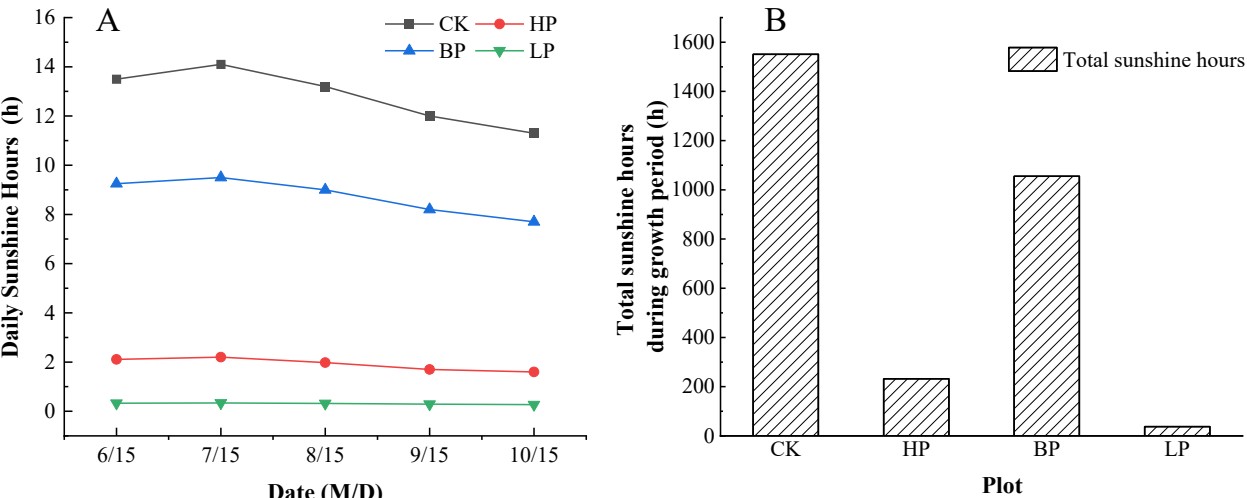

**Figure 4.** The daily sunshine hours (**A**) and total sunshine hours during the rice growth period (**B**). The plot labelled CK was the control plot, which was the not shaded area of the rice plot. LP was the area directly beneath the low edge of the APV panels to the bases of the panels. BP was the inter-row space between the southern and northern APV panels. HP was the area directly beneath the high edge of the APV panels to the bases of the panels.

To clearly compare the differences in total sunshine hours during the growth and development period of rice among different plots, the areas under the four curves in Figure 4A to the x-axis were calculated by integration analysis. The integration results are shown in Figure 4B. The CK plot had the highest total sunshine hours, with a value of 1551.0 h. There were distinct differences in total sunshine hours among the plots in the APV system. The LP plot had the lowest total sunshine hours, with a value of 37.5 h. The total sunshine hours for the HP plot were higher than for LP, reaching 232.1 h. The BP plot received 1055.3 h of total sunshine, which was 68.1% of the CK plot.

To further quantify the effect of PV array deployment on crop photosynthesis, changes in the PPFD of different planting treatment plots in the PV array area were compared under three weather conditions: sunny, cloudy, and rainy days during the rice growth period. The photonic fluxes during one day of typical weather conditions during rice growth are shown in Figure 5. Under three distinct weather conditions, PPFD at 6:00–18:00 time points during the daytime exhibited a single-peak curve, which increased with the rise in the solar altitude angle from 8:00 to 14:00, and subsequently decreased after reaching its peak at 14:00 to 15:00, thereby reflecting the daily variation pattern of solar radiation. The data indicate significant fluctuations on days with high levels of sunlight, less pronounced variations on days with precipitation, and intermediate levels on overcast days. This suggests a weakening effect of the cloud cover on light energy.

During clear weather conditions, the light characteristics of the PV array area exhibited variations, such as oblique shadow light, partially shaded areas, light leakage from the edges of the PV panels, direct light shaded areas, and shadows that varied with the movement of the sun. The shadow area reaches its maximum at midday, after which shadows begin to shorten. The oblique light is subsequently shaded again, and by sunset, the area becomes completely shadowed, resulting in a shading rate of 30–80%. The total PPFD values in the BP plot were 660, 2360, and 8770 $\mu mol/(m^2 \cdot s)$ on rainy, cloudy, and sunny days, respectively. The reduction percentage of PPFD compared with CK was 50.7%, 40.2%, and 10.1% on rainy, cloudy, and sunny days. The proportion of PPFD on cloudy days

weakened by PV panel shading is higher than that on sunny days, and the contribution of diffuse light reflection between PV panels is significant. During rain, the relative PPFD reduction peaked (50.7%), yet the absolute shading effect diminished as ambient light dropped to 10%~20% of sunny conditions.

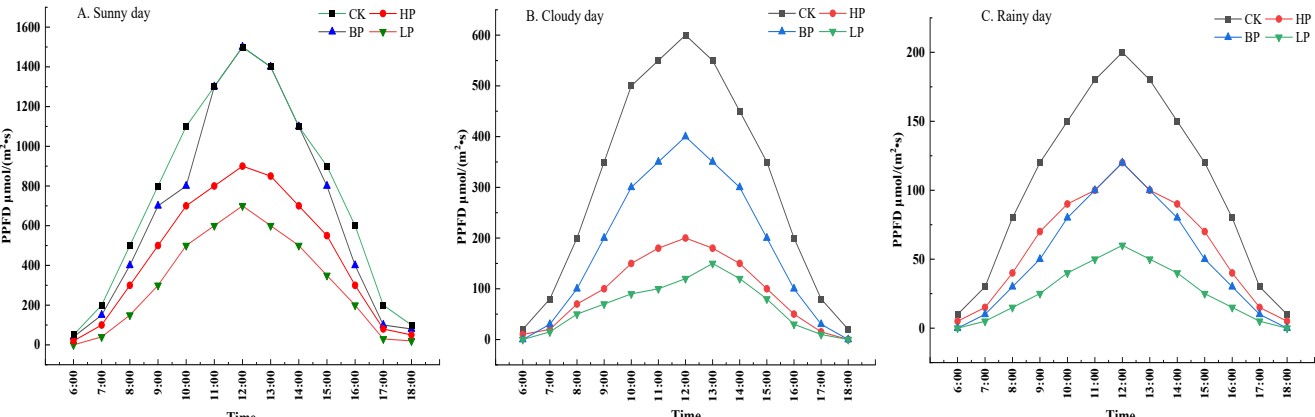

**Figure 5.** Daily change in PPFD under different weather conditions in APV area ((**A**) sunny day, (**B**) cloudy day, (**C**) rainy day). The plot labelled CK was the control plot, which was the not shaded area of the rice plot. LP was the area directly beneath the low edge of the APV panels to the bases of the panels. BP was the inter-row space between the southern and northern APV panels. HP was the area directly beneath the high edge of the APV panels to the bases of the panels.

### 3.3. Rice Growth in the APV System

The alterations in rice plant height during the 2023 and 2024 growing seasons are shown in Figure 6. The deployment of PV arrays substantially affected the height of the rice plants. The results indicated that the rice plant height for each treatment increased with the advancement of the fertility period. At the seedling stage, no significant differences were observed between treatments. However, at the end of the tillering stage, the rice plant height increased rapidly, and significant differences were observed between the plant height of the treated and control plants. Nevertheless, the height of the treated plants was significantly lower than that of the control plants and significantly higher than that of the plants following the panels. A substantial disparity in the stature of the rice plants was observed among the four treatments during the stages of booting and spikelet. The decrease percentages were 14.5%, 22.0%, and 26.8% for the BP, HP, and LP plots, respectively. At the tasselling and grouting stages, the control group exhibited the maximum height, which was significantly higher than that observed in the APV array area. The rice plants in the control group had the greatest height, which was significantly higher than that of the plants grown in the APV array area. During the four periods of rice growth, the plant heights were ranked in the following order: CK> BP > LP > HP. At maturity, the height of the rice plants in BP was approximately 85.4% of the height of CK.

The number of stem sprouts at the end of tillering directly correlated with the number of effective spikes and the final yield of rice. The incidence of stem damage at the conclusion of tillering in rice plants exhibited substantial variation among the various treatments, attributable to the effect of shading caused by APV arrays on the light environment conducive to rice growth. The number of stem sprout incidents at the end of tillering in rice plants during 2023 and 2024 is listed in Table 3. The number of stem sprouts in rice was ranked in the following order: CK > BP > LP > HP. In the control group, the number was significantly higher than that in the LP, BP, and HP treatment groups under APV shading and was 1.18-fold higher than that in BP. No significant difference was found between LP and BP, although both had significantly more sprouts than HP.

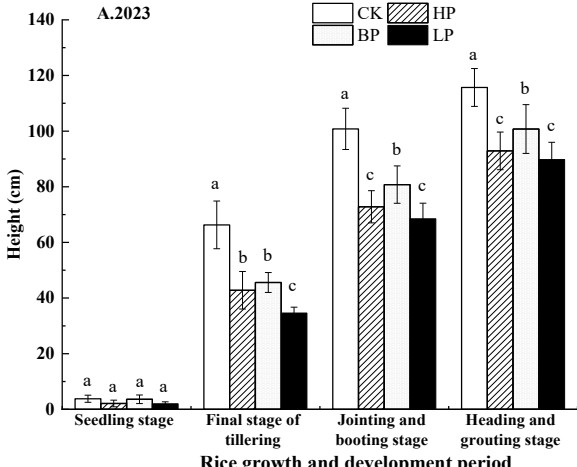 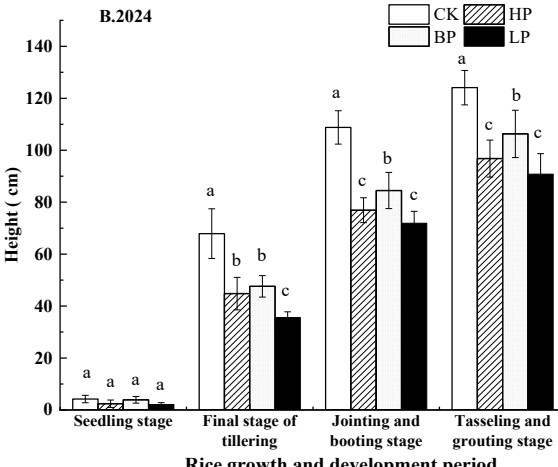

**Figure 6.** Changes in height during the growth and development of rice in the APV system in 2023 (**A**) and 2024 (**B**). The values represent the means of each treatment, and the error bars indicate the standard deviation. Different letters indicate significant differences at *p* < 0.05. The plot labelled CK was the control plot, which was the not shaded area of the rice plot. LP was the area directly beneath the low edge of the APV panels to the bases of the panels. BP was the inter-row space between the southern and northern APV panels. HP was the area directly beneath the high edge of the APV panels to the bases of the panels.

**Table 3.** The number of stem tillers (mean ± SD) in rice at the end of tillering across different cropping plots in the APV system.

| Year | Treatments | | | |
|------|-----|-----|-----|-----|
| | CK | LP | BP | HP |
| 2023 | 9.11 ± 0.99 a | 5.73 ± 0.95 c | 7.83 ± 1.32 b | 7.22 ± 1.14 b |
| 2024 | 9.42 ± 1.12 a | 5.49 ± 0.88 c | 8.97 ± 1.28 b | 7.34 ± 2.15 b |

**Note:** The different letters in the columns indicate significant differences (*p* < 0.05) between the treatments. The plot labelled CK was the control plot, which was the not shaded area of the rice plot. LP was the area directly beneath the low edge of the APV panels to the bases of the panels. BP was the inter-row space between the southern and northern APV panels. HP was the area directly beneath the high edge of the APV panels to the bases of the panels.

### 3.4. The Photosynthesis Parameters of the Expanded Rice Leaf in the APV System

3.4.1. Changes in SPAD Values in Rice

SPAD values, which serve as a proxy for leaf chlorophyll content, indirectly reflect nitrogen levels in rice and guide field management decisions. The SPAD values of the penultimate rice leaf during the 2023 and 2024 growing seasons are shown in Figure 7. The SPAD values of rice in all treatments exhibited an increase in the progression of the fertility period. Subsequent to the onset of the tassel-filling stage, the SPAD values of rice in the control group exhibited a gradual decline, whereas the SPAD values of rice at various locations within the APV array area continued to demonstrate a modest increase. Compared with the SPAD values of rice in CK, the SPAD values of plants between and under the APV panels decreased by 9.36–46.05% during the entire reproductive period. The SPAD values of rice in each treatment at different reproductive periods were ranked in the order of CK > BP > HP > LP. The SPAD values of the penultimate leaves in the APV array area were significantly decreased.

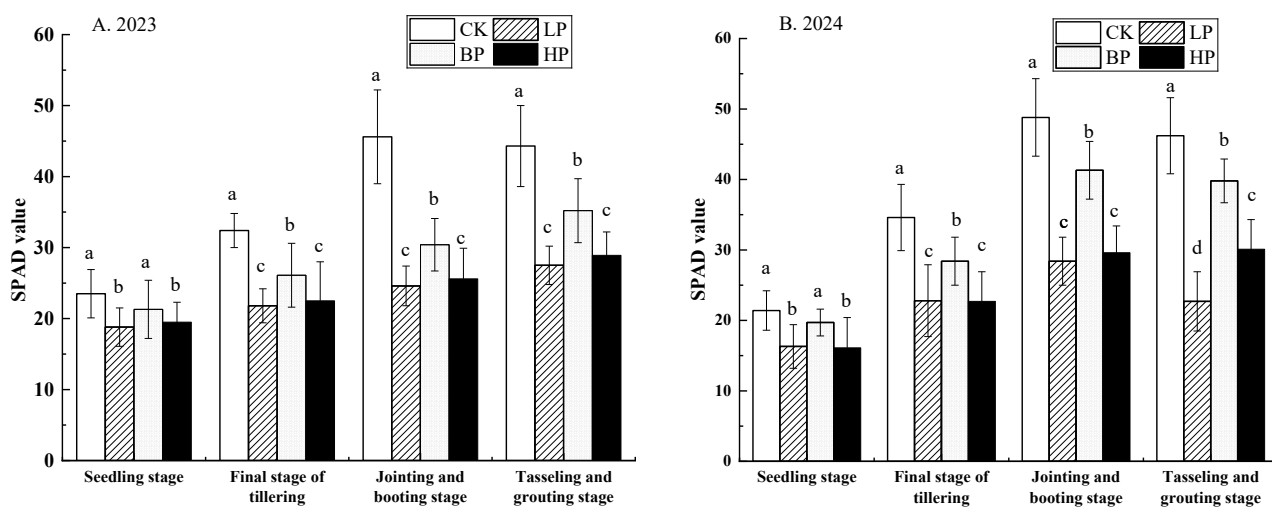

**Figure 7.** SPAD values of penultimate leaves at different growth stages of rice in the APV areas ((**A**) 2023, (**B**) 2024). The values represent the mean of each treatment, and the error bars indicate the standard deviation. Different letters indicate significant differences at $p < 0.05$. The plot labelled CK was the control plot, which was the not shaded area of the rice plot. LP was the area directly beneath the low edge of the APV panels to the bases of the panels. BP was the inter-row space between the southern and northern APV panels. HP was the area directly beneath the high edge of the APV panels to the bases of the panels.

3.4.2. Daily Variations in Photosynthetic Parameters in Penultimate Rice Leaves

The penultimate leaf plays a key functional role during the rice grain filling period, and its photosynthetic capacity directly affects the degree of grain filling and final yield. The photosynthetic characteristics reflect the light energy utilisation efficiency of the lower and middle canopy leaves and provide a key physiological basis for informed rice management. The daily changes in the net photosynthesis rate (Pn) and transpiration rate (Tr) of the penultimate leaf during the full bloom period to the beginning of the filling period during a sunny day are listed in Table 4. Pn and Tr of the penultimate leaf in different planting areas of the APV system exhibited significant variations. In 2023, the photosynthetic efficiency was optimal in BP, which was close to 77% of CK and exhibited a significant reduction compared with that of CK ($p < 0.05$). The Pn values for HP and LP were significantly lower than those of BP, indicating a significant reduction in shade (Pn was only 30–40% of the CK). The trend of Tr was consistent with that of Pn, and Tr was maintained at a high level in the BP plot of the APV systems, although they were significantly lower than that of CK ($p < 0.05$). The dynamics were the same for Pn and Tr in 2024.

Daily variations in the stomatal conductance (Gs) and intercellular $CO_2$ concentration (Ci) for the penultimate rice leaf during sunny days from the bloom to the early filling stages of rice are listed in Table 5. The variation tendencies were the same for Gs and Ci in 2023 and 2024. In 2024, the Gs value was at its peak in the BP plot, up to 0.40 mmol m$^{-2}$ s$^{-1}$ at 10:00, which was significantly higher than that of CK (0.35). Conversely, it was at its lowest in LP (only 0.15 at 12:00), which indicates that intense shading severely impeded stomatal function. Gs decreased from 10:00 to 12:00 in all groups, which was 20% of CK, and recovered slightly in the late afternoon (14:00). The Ci of LP was consistently the highest. The value reached 350 μmol m$^{-2}$ s$^{-1}$ at 12:00, which was 35% higher than that of CK, suggesting that shading significantly reduced the efficiency of $CO_2$ assimilation efficiency. The Ci of BP was close to that of the control. The maximum Ci was observed at 12:00 for all plots (photoinhibition was most pronounced) and decreased at 16:00 with decreasing light intensity. In general, significant differences in Gs were observed among the four plots

across the four time points. No significant differences in Ci were observed between CK and BP at any time point, whereas significant differences were observed between LP and HP.

**Table 4.** Daily changes in Pn and Tr in the penultimate rice leaves in the APV system.

| Year | Time | Pn ($\mu mol \cdot m^{-2} \cdot s^{-1}$) | | | | Tr ($\mu mol \cdot m^{-2} \cdot s^{-1}$) | | | |
|---|---|---|---|---|---|---|---|---|---|
| | | CK | LP | BP | HP | CK | LP | BP | HP |
| 2023 | 10:00 | 18.51 ± 2.12 a | 8.82 ± 1.53 c | 11.62 ± 0.81 b | 8.93 ± 0.71 c | 4.81 ± 0.54 a | 2.04 ± 0.24 d | 4.01 ± 0.41 b | 3.24 ± 0.31 c |
| | 12:00 | 22.29 ± 1.83 a | 7.43 ± 2.02 c | 17.21 ± 1.42 b | 7.24 ± 0.52 c | 6.23 ± 0.62 a | 2.41 ± 0.31 d | 5.03 ± 0.52 b | 3.81 ± 0.42 c |
| | 14:00 | 16.74 ± 2.01 a | 6.61 ± 1.21 c | 14.83 ± 1.23 b | 6.91 ± 0.73 c | 5.54 ± 0.71 a | 2.23 ± 0.23 d | 4.64 ± 0.61 b | 3.52 ± 0.31 c |
| | 16:00 | 12.43 ± 1.52 a | 5.14 ± 1.80 d | 10.34 ± 1.31 b | 6.52 ± 0.93 c | 3.93 ± 0.44 a | 1.83 ± 0.24 c | 3.52 ± 0.34 a | 2.83 ± 0.21 b |
| 2024 | 10:00 | 22.47 ± 2.05 a | 10.33 ± 1.62 c | 14.28 ± 0.92 b | 9.75 ± 0.68 c | 4.92 ± 0.56 a | 2.11 ± 0.25 d | 4.17 ± 0.43 b | 3.35 ± 0.32 c |
| | 12:00 | 24.17 ± 1.75 a | 6.89 ± 1.98 c | 18.36 ± 1.38 b | 6.75 ± 0.49 c | 6.47 ± 0.65 a | 2.28 ± 0.29 d | 5.24 ± 0.54 b | 3.95 ± 0.43 c |
| | 14:00 | 17.82 ± 2.10 a | 6.79 ± 1.22 c | 13.45 ± 1.20 b | 7.05 ± 0.75 c | 5.43 ± 0.69 a | 2.31 ± 0.24 d | 4.52 ± 0.59 b | 3.45 ± 0.30 c |
| | 16:00 | 13.27 ± 1.58 a | 4.87 ± 1.76 d | 11.05 ± 1.35 b | 6.93 ± 0.95 c | 4.02 ± 0.45 a | 1.89 ± 0.25 c | 3.48 ± 0.33 a | 2.75 ± 0.20 b |

**Note:** The different letters in the table columns indicate significant differences ($p < 0.05$) between the treatments. The plot labelled CK was the control plot, which was the not shaded area of the rice plot. LP was the area directly beneath the low edge of the APV panels to the bases of the panels. BP was the inter-row space between the southern and northern APV panels. HP was the area directly beneath the high edge of the APV panels to the bases of the panels.

**Table 5.** Daily changes in Gs and Ci for the penultimate rice leaves in the APV system.

| Year | Time | Gs $mmol \cdot m^{-2} \cdot s^{-1}$ | | | | Ci $\mu mol \cdot m^{-2} \cdot s^{-1}$ | | | |
|---|---|---|---|---|---|---|---|---|---|
| | | CK | LP | BP | HP | CK | LP | BP | HP |
| 2023 | 10:00 | 0.34 ± 0.05 ab | 0.19 ± 0.03 c | 0.39 ± 0.06 a | 0.29 ± 0.04 b | 245.33 ± 20.22 c | 325.81 ± 41.02 a | 252.74 ± 25.41 c | 275.39 ± 29.84 b |
| | 12:00 | 0.33 ± 0.05 a | 0.15 ± 0.03 c | 0.27 ± 0.04 b | 0.23 ± 0.03 b | 269.48 ± 20.34 c | 349.15 ± 45.18 a | 263.25 ± 24.92 c | 307.92 ± 34.87 b |
| | 14:00 | 0.32 ± 0.06 b | 0.15 ± 0.02 c | 0.36 ± 0.04 a | 0.29 ± 0.05 b | 263.17 ± 24.34 c | 328.45 ± 39.85 a | 255.33 ± 29.15 c | 288.72 ± 29.12 b |
| | 16:00 | 0.25 ± 0.03 b | 0.23 ± 0.02 b | 0.31 ± 0.04 a | 0.24 ± 0.03 b | 235.33 ± 15.22 c | 295.81 ± 36.02 a | 242.74 ± 20.41 c | 265.39 ± 26.87 b |
| 2024 | 10:00 | 0.35 ± 0.05 a | 0.18 ± 0.03 c | 0.40 ± 0.06 a | 0.30 ± 0.04 b | 240.15 ± 20.27 c | 320.34 ± 40.14 a | 250.34 ± 25.22 c | 280.51 ± 30.87 b |
| | 12:00 | 0.28 ± 0.04 b | 0.15 ± 0.03 c | 0.35 ± 0.05 a | 0.25 ± 0.03 b | 260.67 ± 25.15 c | 350.64 ± 45.54 a | 270.85 ± 20.28 c | 310.62 ± 35.33 b |
| | 14:00 | 0.32 ± 0.06 b | 0.17 ± 0.02 c | 0.38 ± 0.04 a | 0.28 ± 0.05 b | 250.19 ± 30.34 c | 33028 ± 40.37 a | 260.64 ± 25.57 c | 290.91 ± 30.18 b |
| | 16:00 | 0.25 ± 0.03 b | 0.22 ± 0.02 b | 0.30 ± 0.04 a | 0.22 ± 0.03 b | 230.22 ± 15.17 c | 300.22 ± 35.19 a | 240.61 ± 20.75 c | 270.34 ± 25.28 b |

**Note:** The different letters in the table columns indicate significant differences ($p < 0.05$) between the treatments. The plot labelled CK was the control plot, which was the not shaded area of the rice plot. LP was the area directly beneath the low edge of the APV panels to the bases of the panels. BP was the inter-row space between the southern and northern APV panels. HP was the area directly beneath the high edge of the APV panels to the bases of the panels.

*3.5. Changes in Dry Matter Accumulation and Rice Yield in Different Planting Plots of APV Arrays*

The yield and composition of rice from different planting plots under the APV system are listed in Table 6. Compared with the control, rice in the APV system exhibited a clear yield reduction trend. LP treatment resulted in the greatest yield reduction, with a two-year average decrease of 53.27%. In contrast, BP treatment exhibited the least reduction, with an average decrease of 18.08%. The HP treatment group exhibited an average yield reduction of 41.18%. The yield indexes of the four plots in 2024 showed slight improvement compared with 2023, whereas the pattern of differences between the treatments remained consistent.

**Table 6.** Yield and composition factors of rice in different plots in APV system.

| Year | Plot | Effective Spikes (No/m²) | Fruit Rate (%) | Thousand Kernel Weight (g) | Grain Weight per Spike (g) | Yield (kg/m²) | Yield Reduction Rate (%) |
|---|---|---|---|---|---|---|---|
| 2023 | CK | 241.19 ± 4.27 a | 84.36 | 26.44 ± 1.54 a | 4.13 ± 0.21a | 0.98 ± 0.19 a | --- |
| | LP | 155.78 ± 4.34 c | 49.34 | 24.85 ± 0.88 c | 2.26 ± 0.19a | 0.46 ± 0.14 a | 53.06 |
| | BP | 188.24 ± 4.71 b | 78.15 | 26.13 ± 1.37 a | 3.93 ± 0.20a | 0.81 ± 0.16a b | 17.34 |
| | HP | 163.85 ± 4.67 c | 61.43 | 25.37 ± 1.42 b | 2.97 ± 0.22a | 0.57 ± 0.15a b | 41.18 |
| 2024 | CK | 249.36 ± 4.36 a | 86.78 | 27.11 ± 1.51 a | 4.18 ± 0.15a | 1.01 ± 0.17 a | --- |
| | LP | 157.34 ± 5.10 c | 50.12 | 24.91 ± 1.22 a | 2.33 ± 0.17c | 0.47 ± 0.14 b | 53.47 |
| | BP | 192.16 ± 4.18 b | 78.34 | 27.06 ± 0.97 a | 3.98 ± 0.14a | 0.82 ± 0.11 ab | 18.81 |
| | HP | 165.65 ± 5.24 c | 62.28 | 25.41 ± 1.07 a | 3.01 ± 0.12b | 0.58 ± 0.15 b | 42.57 |

**Note:** The different letters in the table columns in the same year indicate significant differences ($p < 0.05$) between the plots. The plot labelled CK was the control plot, which was the not shaded area of the rice plot. LP was the area directly beneath the low edge of the APV panels to the bases of the panels. BP was the inter-row space between the southern and northern APV panels. HP was the area directly beneath the high edge of the APV panels to the bases of the panels.

Based on ANOVA, compared with CK, there was a significant reduction in the effective spikes under the APV system in 2023 and 2024. There was no significant decrease in the

thousand kernel weight between CK and BP in 2023 and 2024. There were significant differences between BP and HP, BP and LP, and HP and LP in thousand kernel weight in 2023; however, there were no significant differences among the four plots in thousand kernel weight in 2024. There were also no significant differences between CK, BP, and HP in yield, which were significantly higher compared with that of LP.

With respect to the components of rice yield, the effective spike, fruiting rate, thousand grain weight, and spike weight of rice under the APV array were lower compared with that of the control. The fruiting rate exhibited a gradient change in CK > BP > HP > LP. The thousand grain weight of rice in different planting plots under the APV system was reduced by 0.28–1.26 g compared with that of CK, whereas the largest reduction was observed in the planting plots in LP. The spike weight exhibited a trend similar to that of the yield change, and the reduction in treatment in LP was as much as 45.3%.

*3.6. Analysis of Land Use Efficiency in APV Array Area*

Table 6 lists the rice yields of different planting plots in the APV arrays area. According to the characteristics of the APV array layout and treatment design in the field area, the yields were 26.7%, 26.7%, and 46.7% for the HP plot, LP plot, and BP plot separately in the APV system area. Furthermore, we calculated that the APV rice yields were 6370.4 kg/ha and 6627.0 kg/ha in 2023 and 2024. The CK rice yields were 9800.6 kg/ha and 10,100.6 kg/ha in 2023 and 2024. The rice yield under agricultural and PV complementarity was 65–66% of the control rice yield. Combining this with a system efficiency of 82.5% for the PV power plant described by Wenjing et al. [14] resulted in an LER under agricultural and PV complementarity of 148–149%, thus confirming the feasibility of this scheme.

## 4. Discussion

*4.1. Effects of APV Arrays on Sunlight Duration*

Although APV systems have similar climate, eutrophication, air quality, and resource consumption as PV systems [17], the deployment of agrivoltaic arrays significantly alters the duration of direct sunlight for crops, as indicated by the reduction in daily sunshine hours observed across the different planting zones (Table 2 and Figure 4). Compared with CK, HP and LP experienced drastic decreases of 84.37% and 97.56% in sunlight hours, respectively, whereas the inter-panel zone showed a moderate reduction (10.80%). These results are consistent with previous studies demonstrating that PV panel shading redistributes solar radiation, creating heterogeneous microclimates that affect crop growth [10,18]. The pronounced decrease in sunlight availability near the panels underscores the spatial variability of shading effects, which are most severe in zones where shadows persist longest because of low solar altitude angles (early morning and late afternoon).

The diurnal variation in the shadow length (Figure 3) further reveals the dynamic interplay between solar geometry and the PV array configuration. The fitted curves indicate that shadow elongation follows predictable patterns that are associated with solar azimuth intervals, which corroborates the findings of Amaducci et al. [19], who emphasised the role of panel orientation in modulating light regimes. They also coincide with the results of Nasukawa et al. [20], who found a lower PPFD and a reduction in rice yield using the AV system. Notably, the PPFD data (Figure 5) highlight how agrivoltaic systems shift the balance between direct and diffuse radiation. The PPFD is dominated by scattered light from clouds on cloudy days, with no direct light peaks, and is approximately 30–50% of that on sunny days. Cloudy days exhibited less pronounced shading effects because of the dominant diffuse light, a phenomenon consistent with that reported by Trommsdorff et al. [16]. Interestingly, Widmer et al. proposed an optimal daily light integral for crop species, which would be friendly to light-dependent crops [21]. These insights under-

score the need for optimised agrivoltaic system designs that mitigate shading extremes, while maintaining energy output, as proposed in dual-use agrivoltaic frameworks [22]. Future agrivoltaic systems utilising spectrally selective PV materials (e.g., organic and dye-sensitised cells) will enable concurrent crop growth and power generation, thus increasing land productivity by >60% through dual solar energy conversion [23]. Moreover, green-light wavelength-selective organic solar cells were developed for greenhouses, which showed no interference with strawberry and potential in tomato growth [24].

### 4.2. Effects of Agrivoltaic Arrays on Photosynthetic Properties of Crops

The agrivoltaic array significantly influences the photosynthetic properties of rice, as evidenced by reduced SPAD values and altered photosynthetic parameters under shaded conditions (Figure 7; Tables 4 and 5). The decrease in SPAD values (9.36–46.05%) in the APV zones compared with those of the control (CK) suggests that shading reduces the chlorophyll content, which is likely due to lower light availability, limiting nitrogen assimilation, a phenomenon consistent with the results of Reher et al. [25] in beet and wheat-panel systems. However, in a rice yield study in Japan, Ruth et al. [26] showed higher SPAD values in the shaded area, which was different from our results. The reason may be the difference in SPAD values between the flag leaf and the penultimate leaf from the bottom of the rice plant or the growth time being prolonged from shading. Nevertheless, comparing and determining the proper SPAD threshold will be helpful for adjusting the nitrogen fertiliser application. Notably, the BP plot showed a higher Pn (Pn: 77% of CK) than the LP or HP plots (30–40% of CK). The results were consistent with those of Marrou et al. [10], who reported that partial shading preserves photosynthetic efficiency compared with darker shade. Weselek et al. [27] reported that photosynthetic active radiation was reduced by approximately 30% on average under PV. The increased Gs in BP (Table 6) further indicates adaptive stomatal regulation under moderate shade, whereas severe shading (LP) suppressed Gs and increased Ci, reflecting impaired carboxylation efficiency, similar to the observations in shaded maize by Trommsdorff et al. [16]. The lower photosynthesis rate in the APV shading may indicate the adaptation of plants to lower solar radiation [28].

The temporal dynamics of photosynthesis (e.g., midday photoinhibition in all treatments; Tables 4 and 5) underscore the interplay between AV shading and diurnal light stress. The resilience of the BP zone, maintaining higher Pn and Tr, may result from optimised diffuse light capture, as proposed by Weselek et al. [27] for agrovoltaic crops. However, persistent Ci accumulation in the LP zones highlights the chronic light limitation, which corroborates the study of Dinesh and Pearce [18], who noted that prolonged shade disrupts carbon fixation. These results emphasise the need for agrivoltaic designs that balance energy production with the physiological needs of crops, potentially through dynamic panel spacing or spectral-filtering technologies [19].

### 4.3. Effects of AV Arrays on Growth and Yield of Crops

The APV system significantly affected rice growth and yield, with spatial variations in shading intensity leading to differential responses across planting zones (Figure 6; Table 6). The observed reduction in plant height (BP: 85.4% of CK) and tiller number (BP: ~80% of CK) under moderate shading is consistent with previous findings that shading delays vegetative growth but may not proportionally reduce yield [10,29,30]. However, severe shading (LP) caused drastic yield losses (53.27%), primarily because of reduced effective panicles (35% fewer than CK) and lower grain filling (49–50% vs. 84–87% in CK), which is consistent with reports by Dal et al. [31] in wheat-panel systems. Notably, the BP zone maintained relatively stable yields (18% reduction), suggesting that inter-panel spacing enables sufficient light for compensatory growth, as proposed by Amaducci et al. [19]. Because each crop variety

exhibits unique light saturation and shade tolerance for growth [32], it is essential to choose adaptable crop species and the optimal PV array in the APV systems.

Crop yield is determined by a combination of genetic, management, and environmental factors. In the present study, the yield component analysis revealed that APV shading disproportionately affected reproductive traits, reflecting impaired photoassimilation allocation, which is a phenomenon also observed in shaded rice by Thum et al. [33] and Lee et al. [13]. This indicates that the photosynthetic efficacy of rice in APV systems was lower than that in CK. The stronger yield decline in the LP versus HP plots (53% vs. 41%) may stem from the combined shade and microclimate effects, such as reduced morning light interception [16,34], which has a negative role on the number of panicles and weight per panicle. These results highlight the need for optimised APV designs that minimise yield penalties, potentially through dynamic panel orientation [22] or the use of shade-tolerant cultivars [18], to enhance synergy between solar energy generation and crop production.

*4.4. LER of AV System*

The LER is a quantitative metric for the reduction in land use. The LER of 148–149% in this APV system demonstrated superior land use efficiency, in which the combined output of rice (65–66% of monoculture yield) and PV generation (82.5% efficiency) exceeds conventional single-use systems. This is consistent with the findings of Dupraz et al. [1], who proposed that an LER > 100% validates the dual-use advantage of APV systems. The results corroborate the findings of global studies showing that APV systems enhance resource-use efficiency through microclimate regulation and light redistribution [9,10,22]. However, the yield trade-off (34–35% reduction) highlights the need for optimised designs that balance energy and crop production, as suggested by Trommsdorff et al. [16] for site-specific APV implementations in Germany, which are even more advantageous under extreme conditions, such as drought. Consequently, the average grain yield of rain-fed maize was higher and more stable under the agrivoltaic system than under full sunlight in Northern Italy, indicating that AV systems increased crop resilience to climate change [19].

## 5. Conclusions

This study reveals that APV systems create spatially varied light conditions that affect the growth of rice. Although shading reduced sunlight exposure by 10.8–97.6%, between-panel zones depended on 68.1% of sunshine hours (compared with CK), which maintained 77% photosynthetic efficiency and only an 18% yield reduction. However, the heavily shaded areas exhibited 46% lower SPAD values, 57% reduced stomatal function, and 53% yield loss. The system achieved excellent land use efficiency (LER: 148–149%), confirming the potential of APV for sustainable intensification. In the future, experiments on larger areas over more years using photovoltaics panel materials beneficial to plant photosynthesis and various crop species are needed to provide more sustainable solutions and increase the efficiency of APV systems.

**Author Contributions:** Conceptualization, Y.J. and X.G.; methodology, J.L., P.S. and X.S.; investigation, Y.J., J.H. and X.G.; formal analysis, J.H.; data curation, J.L. and X.S.; writing—original draft preparation, Y.J. and X.G.; project administration, P.S. All authors have read and agreed to the published version of the manuscript.

**Funding:** This research was funded by the Natural Science Foundation of Shanxi Province (Grant number 202203021221075); The Natural Science Foundation of Shanxi Province (Grant number 20210302124370); and the Enterprise Research Project (Grant number RH 23000003304 and RH 2400001631).

**Data Availability Statement:** Data are contained within the article.

**Conflicts of Interest:** The authors declare that they have no conflicts of interest.

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
