# Peer review of "Variations in Solar Radiation and Their Effects on Rice Growth in Agro-Photovoltaics System"

_agronomy, doi:10.3390/agronomy15081975_

Round 1
Reviewer 1 Report
Comments and Suggestions for Authors
In Figure 1, define the acronyms “H-L tem and Pre” in the footer of the figure. The size of Figure 2 is outside the margins (adjust the margins)
The size of Figure 2 is outside the margins (adjust the margins). Define the acronyms BP, HP, LP in the figure footer. You should change the color on the left side for better definition. (Yellow? Blue? Choose a different color for better definition.)
The size of Tables 1, 2 and 3 are outside the margins (adjust the margins).
The size of Figure 4 is outside the margins (adjust the margins). Define the acronyms CK, BP, HP, LP in the footer of Figure 4
The size of Figure 5 is outside the margins (adjust the margins). Define the acronyms CK, BP, HP, LP in the figure footer. In A 2023 (Jointing and tasseling stage), I think HP = c; LP = c; check the letters)
Table 4. Define the acronyms CK, BP, HP, LP in the table footer.
The size of Figure 6 is outside the margins (adjust the margins). Define the acronyms CK, BP, HP, LP in the footer of Figure 6
The size of Tables 5, 6 and 7 are outside the margins (adjust the margins).
Table 5. Define the acronyms CK, BP, HP, LP in the table footer.
Author Response
Comments 1: [In Figure 1, define the acronyms “H-L tem and Pre” in the footer of the figure. The size of Figure 2 is outside the margins (adjust the margins)]
Response 1: [Thank you for pointing this out. We added the definition of H-L tem between and Pre as the note parts in figure footer of Figure 1. We adjusted the size of Figure 2 and made sure to show the full scale of Figure 2.]
Comments 2: [The size of Figure 2 is outside the margins (adjust the margins). Define the acronyms BP, HP, LP in the figure footer. You should change the color on the left side for better definition. (Yellow? Blue? Choose a different color for better definition.)]
Response 2: [Thank you for pointing this out. We agreed with it, so we adjusted the display spacing of Figure 2. We added the acronyms of BP, HP, LP in the figure footer of Figure 2. In order to show the plots name clearly, we change the textbox fill color of the left side of Figure 2.]
Comments 3: [The size of Tables 1, 2 and 3 are outside the margins (adjust the margins).]
Response 3: [Thank you for pointing this out. We adjust the margins of the three tables to display them fully.]
Comments 4: [The size of Figure 4 is outside the margins (adjust the margins). Define the acronyms CK, BP, HP, LP in the footer of Figure 4.]
Response 4: [Thank you very much for your good suggestion. We adjusted the size of Figure 4 and added the description of CK, BP, HP, LP as the footer of Figure 4.]
Comments 5: [The size of Figure 5 is outside the margins (adjust the margins). Define the acronyms CK, BP, HP, LP in the figure footer. In A 2023 (Jointing and tasseling stage), I think HP = c; LP = c; check the letters)]
Response 5: [Thank you for pointing this out. We have adjusted the size of Figure 5 to display fully, added the plot name definition as note in the figure footer. We checked the letters of significance difference of HP and LP, corrected the wrong d to c in LP group, there were no significant difference between HP and LP plot.]
Comments 6: [Table 4. Define the acronyms CK, BP, HP, LP in the table footer.]
Response 6: [Thank you for pointing this out. We have added the description of plot names as the table footer in Table 4.]
Comments 7: [The size of Figure 6 is outside the margins (adjust the margins). Define the acronyms CK, BP, HP, LP in the footer of Figure 6.]
Response 7: [Thank you for pointing this out. We adjusted the display size of Figure 6, added the acronyms of CK, BP, HP, LP in the footer of Figure 6. as note. Combined with other reviewers’ comments, we deleted some figures, so the figure numbers were changed.]
Comments 8: [The size of Tables 5, 6 and 7 are outside the margins (adjust the margins).]
Response 8: [Thank you for pointing this out. We have adjusted the display size of tables. Combined with other reviewers’ comments, we deleted some tables, so the table numbers were changed.]
Comments 9: [Table 5. Define the acronyms CK, BP, HP, LP in the table footer.]
Response 9: [Thank you for pointing this out. We have added the description of plots name in the table footer of tables. Combined with other reviewers’ comments, we deleted some tables, so the table numbers were changed.]

Reviewer 2 Report
Comments and Suggestions for Authors
In this contribution X. Gao et al. discussed a systematic presentation of two-year study on agriphotovoltaic systems over rice production indicating that, despite some yield reductions under shaded areas, inter-panel zones maintained high photosynthetic efficiency. overall ,the system achieved a high land equivalent ratio , demonstrating the potential of APV for efficient dual land use. This research topics is relevant to present situation and has a broad interest. I recommend this research to publish after minor rivisions as follows:
- Page 1, line 20: In the abstract please add the full form of "SPAD" for broader field of researchers.
- Page 2, line 47: please add these recent researches of alternatives emerging APV technology through "Green-light Wavelength-selective organic solar modules" for non destructive crop yield nature. Y. Ie et al. Materials Today Energy 2024, 45, 101673. and Y. Ie et al. J. Photopolym. Sci. Technol. 2024, 37, 191-195. this will enrich the introduction through current research advancement in APV.
- Page 3, line 73: In the sub heading please correct the small letter and capital letters as they are intermixed in " Study area and Site Conditions"
- Page 3, line 97: In the subheading please correct the spelling of "experiment".
- Page 9, line 276: "The proportion of PPFD that is weakened by PV panel shading is lower than 277 that of sunny days, and the contribution of diffuse light reflection between PV panels is 278 significant". Please clarify this sentence with proper data.
- Please rewrite the conclusion and with proper future direction.
Author Response
Comments 1: [Page 1, line 20: In the abstract please add the full form of "SPAD" for broader field of researchers.]
Response 1: [Thank you for your comments! We added the meaning of SPAD values, which can be used to reflect the relative chlorophyll content in leaf. It is good for broader field of researchers.]
Comments 2: [Page 2, line 47: please add these recent researches of alternatives emerging APV technology through "Green-light Wavelength-selective organic solar modules" for non destructive crop yield nature. Y. Ie et al. Materials Today Energy 2024, 45, 101673. and Y. Ie et al. J. Photopolym. Sci. Technol. 2024, 37, 191-195. this will enrich the introduction through current research advancement in APV.]
Response 2: [Thank you for pointing this out. We appreciated your suggestion. Green-light wavelength-selective organic solar modules are depending green light to convert electricity, and allowing crops to use blue and red light, which are interesting materials for the PV modules. We added this citation in our paper in the Discussion parts and showed the future practical direction of PV modules.]
Comments 3: [Page 3, line 73: In the sub heading please correct the small letter and capital letters as they are intermixed in " Study area and Site Conditions"]
Response 3: [Thank you for your careful reading. We corrected the capitalization as follows: Study Area and Site Conditions. We also corrected other sub headings.]
Comments 4: [Page 3, line 97: In the subheading please correct the spelling of "experiment".]
Response 4: [Thank you for your careful reading. We corrected the spelling.]
Comments 5: [Page 9, line 276: "The proportion of PPFD that is weakened by PV panel shading is lower than that of sunny days, and the contribution of diffuse light reflection between PV panels is significant". Please clarify this sentence with proper data.]
Response 5: [Thank you for pointing this out. In order to clarify the sentence, we calculated the total PPFD, reduction percentage of each plot in different weather. The expression in the context is as follows:” The total PPFD value in BP plot was 660, 2360 and 8770 μmol/(m2·s) on rainy, cloudy and sunny day, respectively. The reduction percentage of PPFD compared with CK was 50.7%, 40.2% and 10.1% on rainy, cloudy and sunny day. The proportion of PPFD on cloudy days weakened by PV panel shading is higher than that on sunny days, and the contribution of diffuse light reflection between PV panels is significant. During rain, the relative PPFD reduction peaked (50.7%), yet the absolute shading effect diminished as ambient light dropped to 10%~20% of sunny conditions.”]
Comments 6: [Please rewrite the conclusion and with proper future direction.]
Response 6: [Thank you for pointing this out. We added the sunshine hours result and future direction in the conclusion, specially for the development of PV panel materials which are helpful to the photosynthesis of plant.]

Reviewer 3 Report
Comments and Suggestions for Authors
I find the proposed agrophotovoltaic (APV) study for rice cultivation very interesting, especially in areas with high energy and food demand. Competition from large-scale solar farms and food production on arable land can be a problem depending on the type of crop. Therefore, I believe it is interesting to study how certain staple crops (rice) respond when their exposure to solar radiation decreases.
In agrovoltaic systems, it is essential to know and choose the crops that can adapt to the number of hours of sunlight they require, as well as an optimal design for the photovoltaic system that maximizes both energy generation and crop yield.
The conclusions and results are consistent with the stated objectives. The trial design is well-conceived.
Below are some questions and suggestions:
"Agrophotovoltaic system" appears in the Keywords and also in the title. Perhaps it would be better to replace it with "agrivoltaic systems" so as not to repeat the keywords that appear in the title.
Additionally, "land equivalence ratio" is misspelled; it should be "land equivalent ratio." Other misspelled terms should also be corrected, such as the one that appears in the title "2.2 APV array and experiment design" to "2.2 APV array and experiment design."
Figure 2 does not show the units corresponding to the plot lengths (HP, LP, and BP). The number appears to be in "mm," while the text expresses it in "m." It would be helpful if they appeared in the same units.
Table 1 shows "Solar altitude angle, azimuth, and effective shadow length on June 15, 2023, in the study site," and Figure 3 represents "Day variation of effective shadow length of PV panels in the APV area." Since the trial runs from June to October, are the results considered the same as those for June 15? Wouldn't the shadow length vary for, say, August 15?
In your results, I don't understand how you achieved a LER with agricultural and photovoltaic complementarity of 1.48-1.49, based on Table 7.
What new information does this study provide compared to others in the literature, conducted with agrivoltaic systems and rice cultivation?
Author Response
Comments 1: ["Agro-photovoltaic system" appears in the Keywords and also in the title. Perhaps it would be better to replace it with "agrivoltaics systems" so as not to repeat the keywords that appear in the title.]
Response 1: [Thank you for your good comments. It is really a good suggestion. We replaced the keywords ‘agro-photovoltaic system’ with ‘agrivoltaic system’ to avoid repetition.]
Comments 2: [Additionally, "land equivalence ratio" is misspelled; it should be "land equivalent ratio." Other misspelled terms should also be corrected, such as the one that appears in the title "2.2 APV array and experiment design" to "2.2 APV array and experiment design."]
Response 2: [Thank you for pointing this out. They were our misspelling. We corrected them in the 2.2 title and the context part.]
Comments 3: [Figure 2 does not show the units corresponding to the plot lengths (HP, LP, and BP). The number appears to be in "mm," while the text expresses it in "m." It would be helpful if they appeared in the same units.]
Response 3: [Thank you very much for your good suggestion. We have added the unit in the graph with “meter”, meanwhile, we described the characteristics of each plot with the same units, added the plot width of each treatment in the paragraph.]
Comments 4: [Table 1 shows "Solar altitude angle, azimuth, and effective shadow length on June 15, 2023, in the study site," and Figure 3 represents "Day variation of effective shadow length of PV panels in the APV area." Since the trial runs from June to October, are the results considered the same as those for June 15? Wouldn't the shadow length vary for, say, August 15?]
Response 4: [Thank you very much for your inspiring questions. Considering the variance of sun movement and season change, in order to clarify the dynamics of daily sunlight hours during the rice growth period, we calculated daily sunlight hours for each plot on 15 July, 15 August, 15 September and 15 October 2023. According to the daily sunshine hours of these 5 typical days, we could get the point and line plot, which showed the dynamic trend of daily sunshine hours during the rice growth period. Then we based on the mathematics meaning of integration, got the area of each plot during the total growth period of rice by the integration method, which could indicate the summation of sunshine hours under the daily sunshine hours curves to the x-axis, showing the total sunshine hours in different plots. The addition figure was showed as below:]
Comments 5: [In your results, I don't understand how you achieved a LER with agricultural and photovoltaic complementarity of 1.48-1.49, based on Table 7.]
Response 5: [Thank you for pointing this out. It is our deficiencies in expression. In order to explain the result of LER clearly, we added specific expression in the section of 3.6 as follows: According to the characteristics of the APV array layout and treatment design in the field area, there were 26.7 %, 26.7 % and 46.7 % of HP plot, LP plot and BP plot separately in the APV system area. Furtherly, we calculated the APV rice yield were 6370.4 kg/ha and 6627.0 kg/ha in 2023 and 2024. The CK rice yield were 9800.6 kg/ha and 10100.6 kg/ha in 2023 and 2024. Moreover, we replaced the LER value with 148-149% instead of 1.48-1.49. It is easier to understand the calculation process of LER.]
Comments 6: [What new information does this study provide compared to others in the literature, conducted with agrivoltaic systems and rice cultivation?]
Response 6: [Thank you for your good recapitulatory question! According to the two consecutive years field experiment, we quantified the solar radiation difference among BP, HP and LP plot in APV system, evaluated their effects on the rice growth and photosynthetic parameters, finally, we analyzed the yield components in specific plot and evaluated the land efficiency of APV system in rice. As we mentioned in the Introduction 3rd paragraph and summarized in Conclusion part. The new information is as follows: PV shading reduced sunlight exposure by 10.8%–97.6%, between-panel zones maintained 77% photosynthetic efficiency and only an 18% yield reduction. However, the heavily shaded areas exhibited 46% lower SPAD values, 57% reduced stomatal function and 53% yield loss. The APV system achieved excellent land use efficiency about 148-149 %.]

Reviewer 4 Report
Comments and Suggestions for Authors
This research provided useful information for use of agrophotovoltaic system in agricultural production. Generally, it is well-written, especially for Introduction and Discussion. However, more detailed information is necessary to provide in Materials and Methods section, and the data presenting in Results section can be improved. The detailed comments and suggestions can be found in attachment. Thanks

Author Response
Comments 1: [Highlight: Line 85: high solar resource availability.]
Response 1: [Thank you for pointing this out. We modified the expression as: high availability of solar resource]
Comments 2: [Figure 1, it is better to add daily light integral if the data were collected, since light is an important parameter in this paper]
Response 2: [Thank you for pointing this out. Considering the complexity of Figure, we did not add the sunshine hours in it, but we added the expression in the 2.1 on the basic conditions as follows: The total sunshine hours in 2023 and 2024 were 1010 hours and 890 hours, respectively.]
Comments 3: [Highlight: Line 100: 28° fixed tilt angle]
Response 3: [Thank you for pointing this out. We modified the expression as: at a fixed tilt angle of 28°]
Comments 4: ["An adjacent plot without APV system (i.e., with full sun) was in control treatment"? three distinct or four distinct micro-environments? Shading conditions or light conditions?]
Response 4: [Thank you for pointing this out. We modified the redundant expression as: “An adjacent APV system with full sun was used as the control plot (CK). This configuration provided four distinct micro-environments for the comparative analysis of crop performance under varying light conditions, which maintaining uniform soil management practices across all treatments.”]
Comments 5: [Explain How the four treatments were distributed? random completer block design?]
Response 5: [Thank you for your rigorous questions on the experiment design. In our four treatment, we designed random block in the direction of width.]
Comments 6: [Highlight: the solar declination (δ), altitude angle (α) and azimuth angle (β); α is the altitude angle, φ is the geographic latitude of the observation site, δ is the solar declination; β is the azimuth angle, α is the altitude angle, φ is the geographic latitude of the observation site and δ is the solar declination.]
Response 6: [Thank you for pointing this out. These expressions were redundant. We deleted them and left the parameters explanation which were mentioned firstly.]
Comments 7: [δ is the solar declination repeat?]
Response 7: [Thank you for pointing this out. We deleted the repeat expression.]
Comments 8: [whick, which?]
Response 8: [Thank you for pointing this out. It is our careless fault. We corrected it as “which”.]
Comments 9: [by my math, there were 60 measurements for each time point. 60 sensors for automatic data collection? or one meter for multiple measurements? if latter, how was the temporal light variation excluded from treatment difference?]
Response 9: [Thank you for pointing this out. As you mentioned, we did not fix the sensors in the field. When we collected the data in typical weather, we chose five fixed point in each plot and set the time range from 6 am to 6 pm. During the measurement period, we did not move the TES-133P.]
Comments 10: [the subsample number seems to be small relative to experimental unit area]
Response 10: [Thank you for pointing this out. We totally agreed with you. In our field experiment area, the length of plot is enough, but the width is limited, we also considered to avoid margin effect. So the subsample number seems to be small. We will pay attention this next time, increase the subsample number to five or ten as possible.]
Comments 11: [since BP plot is 1.7 fold larger than other plots, maybe the sample area should be a little bit bigger]
Response 11: [Thank you for pointing this out. Considering the ANOVA analysis, we need the same number of replication, so we took the same sample area and same subsample numbers. Generally, we took 10 m2 sampling areas as grain yield measurement, considering the 2 m width of LP and HP plot, and avoiding the marginal effect too, we chose 1 m2 area for yield measurement. Maybe we increase the sample number next time.]
Comments 12: [I assume all the data in this section were calculated. Just wonder how they could be validated]
Response 12: [Thank you for pointing this out. Actually, there are several kinds of methods to calculate the solar parameters, and considering our calculation capacity and data access permission, we chose the basic astronomical formula method, which is based on the spherical trigonometry. We calculated them carefully and did not validate the result with other methods. We will remember the issue and pay attention them in future.]
Comments 13: [This figure repeated a part of the Table 1. Suggest to remove it except for the situation where there were some measured data to compare with the calculated data.]
Response 13: [Thank you for your comment. Please see the response 14.]
Comments 14: [Table 2, consider combined to Figure 2]
Response 14: [Thank you for your comment. Considering your Comments 13 and 14, we combined the Table 2 with the Figure 3, then deleted the Table 2 in the context. The new figure was showed below: ]
Comments 15: [observed? or calculated?]
Response 15: [Thank you for pointing this out. Generally, when the light intensity reaches 10 - 20 μmol/(m2 â–ªs), C3 plants begin to carry out net photosynthesis. Basing on the PPFD in the study site, there were less than 10μmol/(m2 â–ªs) at 6 am, so we determine the earliest measurement time as 6 am.]
Comments 16: [how to determine the number of sunshine hours? cannot find it in M&M. Boardroom? What does it mean?]
Response 16: [Thank you for pointing this out. We determine the number of sunshine hours in different plots, which refers to the period when the sun is directly shining. For example, in the BP plot in APV system, compared to the CK without the photovoltaic array blocking, the start time of direct sunlight in the BP plot will be affected by the south-facing photovoltaic panels, resulting in a delay in the direct sunlight duration, and then it will be affected by the shadow produced by the north-facing photovoltaic panels. In the Materials and Methods of 2.3.2, the calculation of L' is used to evaluate the shadow length of the photovoltaic panels in the north-south direction. This shadow length provides effective shading for the crops located in the APV system. Boardroom was our misspelling, we corrected it as PV panels.]
Comments 17: [“The PPFD is dominated by scattered light from clouds on cloudy days, with no direct light peaks, and is approximately 30-50% of that on sunny days”, I assume it is a speculation. May be better put in discussion section.]
Response 17: [Thank you for pointing this out. We totally agreed with you. We moved this sentence to the Discussion 4.1 section paragraph 2 on the PPFD.]
Comments 18: [what is the decrease percentage?]
Response 18: [Thank you for pointing this out. In the paragraph, we added the decrease percentage as follows: The decrease percentages were 14.5%, 22.0% and 26.8% for BP, HP and LP plot respectively.]
Comments 19: [In Table 4, “plot “put it above the four treatments, and change it to treatments]
Response 19: [Thank you for pointing this out. We corrected them in the Table 3. as below:]
Comments 20: [In 3.4.2, why different parameters were presented in 2023 and 2024?]
Response 20: [Thank you for pointing this out. There were the same tendencies of photosynthesis parameters in 2023 and 2024, so we presented different parameters in 2023 and 2024. Considering your comments, we added the Pn and Tr of 2024, Gs and Ci of 2023 respectively in Table 4 and Table 5. It is comprehensive to reflect the difference of photosynthetic parameter.]
Comments 21: [In 3.6, could it be presented in Table 7?]
Response 21: [Thank you for pointing this out. We added the explanation on LER calculation process as follows: According to the characteristics of the APV array layout and treatment design in the field area, there were 26.7 %, 26.7 % and 46.7 % of HP plot, LP plot and BP plot separately in the APV system area. Furtherly, we calculated the APV rice yield were 6370.4 kg/ha and 6627.0 kg/ha in 2023 and 2024. The CK rice yield were 9800.6 kg/ha and 10100.6 kg/ha in 2023 and 2024.]
Comments 22: [citations are rarely found in Results, if there is a discussion section.]
Response 22: [Thank you for pointing this out. We totally agreed with you. This citation provided the important efficiency values of the PV power plant, which is a constituent of land use type of APV system and is necessary to calculate the LER, so we kept it in the results parts.]
